# Mixture of Tokens: Continuous MoE through Cross-Example Aggregation

**Szymon Antoniak** [*]

**Michał Krutul** [*]
IDEAS NCBR
University of Warsaw

**Maciej Pióro**
IDEAS NCBR
Polish Academy of Sciences

**Jakub Krajewski**
IDEAS NCBR
University of Warsaw

**Jan Ludziejewski**
IDEAS NCBR
University of Warsaw

**Kamil Ciebiera**
IDEAS NCBR
University of Warsaw

**Krystian Król**
IDEAS NCBR
University of Warsaw

**Tomasz Odrzygóźdź**
IDEAS NCBR

**Marek Cygan**
University of Warsaw
Nomagic

**Sebastian Jaszczur** [*]
IDEAS NCBR
University of Warsaw

## Abstract

Mixture of Experts (MoE) models based on Transformer architecture are pushing the boundaries of language and vision tasks. The allure of these models lies in their ability to substantially increase the parameter count without a corresponding increase in FLOPs. Most widely adopted MoE models are discontinuous with respect to their parameters - often referred to as *sparse*. At the same time, existing continuous MoE designs either lag behind their sparse counterparts or are incompatible with autoregressive decoding. Motivated by the observation that the adaptation of fully continuous methods has been an overarching trend in Deep Learning, we develop Mixture of Tokens (MoT), a simple, continuous architecture that is capable of scaling the number of parameters similarly to sparse MoE models. Unlike conventional methods, MoT assigns mixtures of tokens from different examples to each expert. This architecture is fully compatible with autoregressive training and generation. Our best models not only achieve a $3\times$ increase in training speed over dense Transformer models in language pretraining but also match the performance of state-of-the-art MoE architectures. Additionally, a close connection between MoT and MoE is demonstrated through a novel technique we call *transition tuning*.

## 1 Introduction

Transformer-based Large Language Models (LLMs) make up one of the most active fields in AI, exhibiting human-level performance across a variety of tasks, including translation, language understanding, reasoning, and code generation [1, 2, 3]. The exorbitant sizes of all state-of-the-art language models are integral to their success, with parameter counts reaching tens or even hundreds of billions. This phenomenon aligns with findings from [4, 5], where the latter suggests that the optimal model size grows proportionally to the available computational budget. Given that hardware efficiency has been steadily increasing over the past decade [6, 7], these findings imply that scaling will continue to be a vital component in training increasingly capable models.

---

[*]Equal contribution. Work done while at IDEAS NCBR and University of Warsaw.
Detailed authors' contributions are listed in Appendix F.

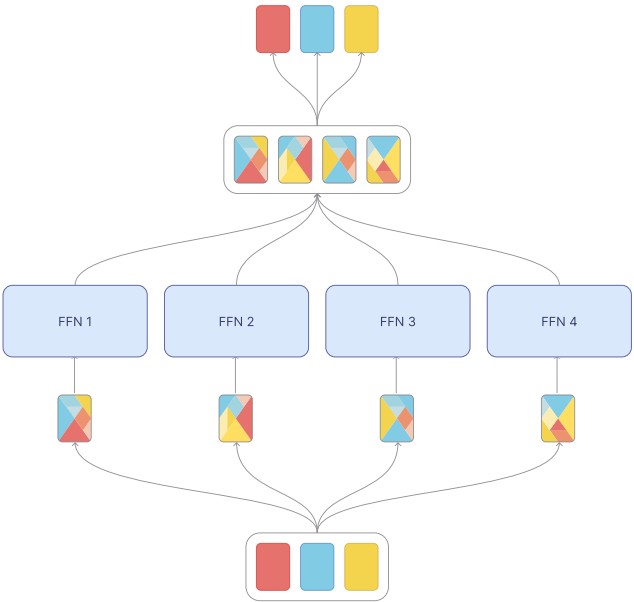

Figure 1: Mixture of Tokens: Each expert receives a unique mixture of tokens in the group. Mixing weights are determined by the controller, which is a fully connected layer (omitted for clarity). For a given token, its update is a linear combination of expert outputs, with the coefficients equal to the token's original mixing weights for each expert.

However, model scaling invariably comes at a cost. Larger models execute more Floating Point Operations (FLOPs) per token, resulting in both training and inference becoming slower and more expensive [8, 9]. Mixture of Experts [10] architectures offer an attractive alternative to standard Transformers by drastically increasing the number of parameters. The core idea is to have multiple *experts*, each specializing in a different part of the input space. Currently, state-of-the-art models based on MoE leverage sparsity by activating only a fraction of the parameters for each token [11]. This allows the networks to increase the number of parameters by an order of magnitude while keeping the FLOPs per token roughly constant. In this work, we use MoE to signify sparse Mixture of Experts architectures unless explicitly stated otherwise.

The aforementioned sparsity is made possible with a *router*, a small network that selects the best experts for each token. This makes the output of an MoE layer discontinuous with respect to its parameters, as only a subset of the experts is chosen for each token (this is typically done with a discrete *top-k* operation). The discontinuity and the resulting fluctuations of the router's decisions have been shown to hurt training efficiency [12, 13] and are hypothesized to be a source of training instability in large MoE models [14, 15]. Conversely, existing continuous MoE architectures involve trade-offs, including the inability to scale [16, 17], or incompatibility with autoregressive decoding [15].

This paper introduces Mixture of Tokens, a novel, continuous Transformer architecture closely related to sparse Mixture of Experts. Similar to MoE, it can support large parameter counts without significant costs in FLOPs. The core idea behind our design is for each expert to process not individual tokens separately, but their combined representation.

This technique results in a continuous model that avoids the top-k operation. It requires no additional techniques commonly required in existing MoE designs (both sparse and continuous), such as load balancing losses, calculating solutions to optimization problems, or non-homogeneous training schedules [17, 18, 12]. It is capable of scaling the parameter counts akin to sparse MoEs and is compatible with autoregressive language modeling and generation. Our analysis demonstrates a $3\times$ speedup over a dense baseline and improved stability over MoE.

In summary, our contributions are the following:

- Introducing the novel Mixture of Tokens (MoT), a continuous Mixture of Experts architecture that mixes tokens from different examples for joint processing.
- An analysis of scaling properties of Mixture of Experts models on multiple scales.
- Introducing transition tuning, allowing a pretrained MoT model to be tuned for sparse MoE inference if desired.

## 2 Background and Related Work

In this section, we provide an overview of the approaches related to our work and discuss the differences between various MoE designs. We will introduce the Mixture of Tokens architecture in Section 3 and provide a detailed comparison between MoT and related methods in Section 3.4.

### 2.1 Large Language Models

Transformer scaling has been shown to be a critical factor in achieving state-of-the-art results in language and vision tasks [4, 19], with the largest disclosed parameter counts in dense models reaching hundreds of billions of parameters [20, 3, 2]. These large models exhibit impressive abilities not present in their smaller counterparts [21]. [4] and [5] have demonstrated that the final model performance is predictable and correlates directly with the model size and the amount of training data. However, increasing model sizes raises the demand for computational resources during both training and inference [22].

### 2.2 Mixture of Experts

Mixture of Experts (MoE) was first introduced by [10, 23] as an ensemble-like neural network comprised of separate sub-networks called experts. The original design used a gating network to select a soft assignment of experts for each input. In the context of Deep Learning, the notion of an MoE *layer* was introduced in [24]. [25] combined a sparse version of MoE layers with an LSTM to train a model with over 100 billion parameters, which was unprecedented at the time. The design, similar to state-of-the-art MoE models today, used a small routing network to decide the top-k best experts for each input. By choosing only a subset of the experts, they were able to increase the size of the network while keeping FLOPs per token roughly constant. The Transformer was first combined with MoE layers in [26], where it replaced the Feed-Forward layer. The design was further simplified in [27], which trained a model with 1.6 trillion parameters using top-1 routing. Since then, a number of studies have investigated different sparse MoE designs [28, 29, 30, 31, 32]. A comprehensive analysis of scaling properties of sparse MoE architectures can be found in [33]. [34] introduced the notion of granularity, which in spirit is similar to the number of groups in MoT, described in Section 3.2.

### 2.3 Continuous Mixture of Experts

Continuous architectures serve an important role within the field due to their flexible and efficient nature. [17] were pioneers in introducing them in MoE by presenting continuous techniques for calculating encodings of the choice of an expert. In another approach, [16] proposed a method in which they merge experts based on the weights of the router network. In a recent advancement, [15] proposed a continuous variant of MoE for the Vision Transformer, where patches are mixed only within each image.

### 2.4 From Hard to Soft Methods

From the very beginning of the Deep Learning field, there has been a shift from discrete functions toward continuous ones. The first perceptron [35] used "all-or-none" activation, supposedly to align with propositional logic. This was later improved with soft activation functions, enabling gradient descent and multi-layer neural networks. Similarly, soft attention, introduced in [36], enabled RNNs to look at arbitrary input from the past while maintaining the ability to learn the selection with standard gradient descent. This is in contrast to hard attention, which requires, e.g., reinforcement

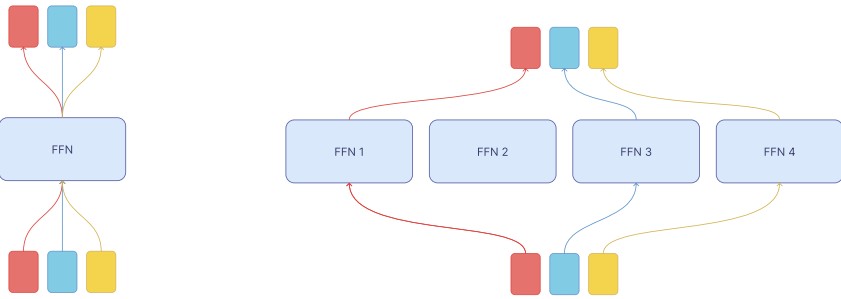

Figure 2: **(Left).** Diagram of a standard Feed-Forward layer featured in the Transformer architecture: each token is processed with the same MLP independently of other tokens. **(Right).** Diagram of a Token Choice layer, where each token decides which expert to choose. In this way, different experts process a different number of tokens. If one expert is chosen by too many tokens, a portion of the tokens is dropped — they receive no update.

learning techniques. While hard attention could perform on par with soft attention [37, 38], soft attention, with its simplicity of training, offered better trade-offs and was later used as the basic building block of the Transformer [39].

Mixture of Experts, introduced into Deep Learning by [10, 23, 25], appears to be inherently a discrete function—after all, the expert either processes a given token or it does not. However, similar to the transition from hard to soft attention, an expert in MoE can "attend" to a combination of tokens, taken as a weighted average. This results in a smooth, continuous model and facilitates more stable training.

## 3    Mixture of Tokens

The goal of this work is to develop an efficient, continuous architecture that retains the scalability of Mixture of Experts, while simultaneously omitting the top-k operation, which limits a token's exposure to different experts. An intuitive approach to achieving this is to route all tokens to all experts, but this is computationally infeasible for large-scale pretraining. To overcome this constraint, the method explored in this work considers what happens not to an individual token but to a whole group of tokens instead. The main contribution of this work is the observation that allowing an expert to dynamically produce a continuous representation of the entire group of tokens, which is more lightweight to process than each token individually, yields positive results.

---
**Algorithm 1** Mixture of Tokens layer

---
1: **for** each $E$ in experts **do**:
2:     $\text{weights}_E = \text{Softmax}(\text{Linear}(\text{tokens}))$
3:     $\text{mix} = \sum_i \text{token}_i * \text{weights}_{i,E}$
4:     $\text{output}_E = E(\text{mix})$
5: **for** each i **do**
6:     **for** each E **do**
7:         $\text{update}_i = \sum_E \text{output}_E * \text{weights}_{i,E}$

---

More specifically, in our design, an input batch is divided into groups of tokens, and each group is processed independently. Given a group and a single expert, a scalar weight is produced for each token. The weights are then normalized and used to compute a linear combination of the tokens, which is used as the expert's input. The experts' outputs are used for token updates as follows: for each input token, its update is a linear combination of expert outputs, with the token's mixing weights for each expert as coefficients[2]. A diagram of our method is presented in Figure 1.

---
[2]The authors note that an MoT layer admits an efficient vectorized implementation, where all meaningful computations are done with batched matrix multiplications.

To see why this method is scalable, it is helpful to examine the relationship between the number of tokens in a group and the number of experts. Essentially, if these two quantities are equal, the total computation performed by the experts is the same as in the case of top-1 routing. This allows MoT to benefit from the same parameter scaling as seen in MoE, which we confirm empirically in Section 4.2.

## 3.1  Intuition Behind Our Method

As we mix tokens from multiple unrelated sequences, we do not expect the model to meaningfully use the information from one sequence to improve prediction in a different sequence. However, we hypothesize that such mixing (1) provides richer feedback (gradients) to train the model, especially the router, and (2) results in a smoother loss landscape, which is resistant to small perturbations in inputs and weights.

Intuitively, for a given expert, from the perspective of each token, the token receives a certain amount of update to its representation (in the residual stream) based on:

- Itself, producing a proper signal expected to improve the token representation.

- Tokens other than itself, which are essentially random tokens from unrelated sequences. As these sequences are randomly sampled from the dataset, the impact from these tokens will point in random directions and, essentially, just add some amount of noise to the token update.

We generally expect neural networks to be resistant to a certain amount of noise added to them. Moreover, while the signal-to-noise ratio worsens for tokens with low expert weight, the expert weight also modulates the magnitude of the update. Therefore, the amount of noise added to the representation is limited.

We stipulate that MoT experts will learn to focus on a single token or a small number of tokens, thereby minimizing noise and approximating sparse MoE when optimal. However, other tokens will be assigned nonzero weight, allowing some information to flow to the router for each and every token-expert pair, unlike sparse MoE. Additionally, the output of MoT is more continuous, with small perturbations of input/weights corresponding to small changes in the output rather than large discrete jumps, as may occur in the case of sparse MoE.

## 3.2  More Mixtures per Expert

Building on the design described above, we experiment with feeding more than one mixture into each expert. Without further modifications, this approach would result in a linear increase in computational costs for each additional mixture processed. To circumvent this added cost, MoT uses more experts, but each expert has a proportionally reduced hidden dimension. This way, each mixture is processed by a smaller expert, and the layer's total number of parameters, as well as the number of FLOPs used by all experts, remains approximately the same. We find that this design consistently improves MoT as the number of processed mixtures increases, just as it does in sparse MoE [34]. Likewise, the optimal granularity aligns roughly with the number of mixtures in this work.

## 3.3  Token Groups in Mixture of Tokens

The question of how token *groups* are decided within a batch is crucial for compatibility with autoregressive training and inference. The main insight here is that tokens from the same sequence cannot be placed in a single group, as the mixing operation would result in an information leak. Due to this restriction, MoT groups tokens from different examples based on their position in the sequence. Thus, all tokens within a group share the same position in their respective sequences. As previously mentioned, to maintain a constant number of FLOPs per token, an increase in the number of experts means an equal increase in group size. An illustration of how grouping is done within a batch of tokens is shown in Figure 3.

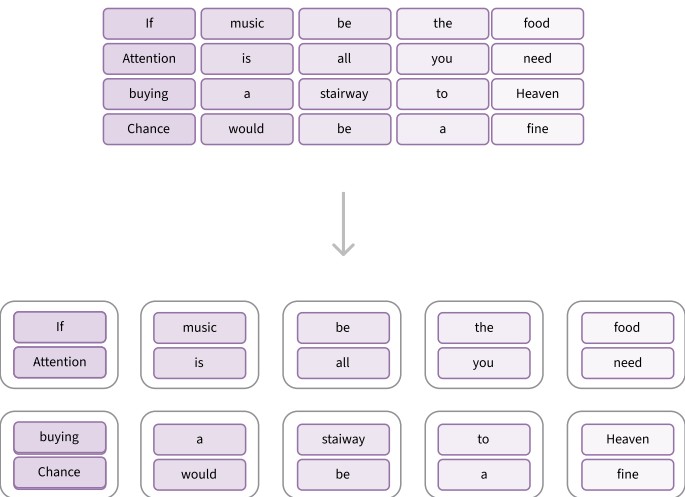

Figure 3: Each group consists of tokens with the same position in a sequence. In this example, the group size is 2. Note that the maximum possible group size equals the size of the batch.

## 3.4 Comparison with Other Mixture of Experts Architectures

**Scaling** The technique featured in [17] is based on a continuously differentiable sparse top-k router, which is a major advantage compared to the common top-k gating. However, this approach requires that all experts are utilized in a portion of training, rendering it computationally prohibitive for models with large numbers of experts. The architecture based on merging experts proposed in [16] also offers an attractive, continuous alternative to top-k gating, yet the cost of merging all experts again scales linearly with the number of experts. To address this, the technique is applied once per sequence, which limits the expressive power of the final model.

**Training stability** [26] reported instabilities during the training of large MoE models, stemming from the inaccuracies when calculating router weights in low precision. To stabilize the training, they resorted to using full precision. [27] made progress in using mixed precision when training MoE by using selective high precision for gating. When comparing [26, 27] to MoT, an advantage of our technique emerges - it is more robust to training in lower precision than other methods. We conjecture, that this is due to the merging mechanism being less susceptible to rounding errors than gating in sparse MoEs.

**Token dropping** Token dropping is a phenomenon where tokens do not receive an update from any expert. This can happen when the expert was selected by too many tokens in a batch [27, 26, 40] or, in the case of routing experts to tokens, when a token is not selected by any expert [30]. Existing techniques to combat this phenomenon offer a partial solution, yet the problem persists. In contrast, tokens in MoT are part of every mixture produced within their group; hence, they always receive an update.

**Auto-regressive decoding** Mixture of Tokens is based on the concept of merging tokens before they are processed by an expert. An encoder-only design of a similar nature is featured in concurrent work [15]. The technique is based on merging patches within an image for vision models, i.e., within a single sample. This should be contrasted with MoT, which merges tokens from different sequences within a batch. This crucial difference allows MoT to be compatible with autoregressive training and inference.

**Time complexity** The time complexity of our approach is identical to that of the Token Choice and Expert Choice methods. In all cases, the cost of computing routing logits is of order $O(d_{model} \cdot N_{experts} \cdot N_{tokens})$.

# 4 Experiments

The focus of this work is to investigate the efficiency of Tokens on autoregressive language modeling. To measure model quality, we pretrain models for a fixed number of tokens and compare final perplexity in accordance with existing MoE literature [28, 27]. In all experiments, the models are trained on the C4 dataset[3] [41] and use the GPT-2 tokenizer. Unless specified otherwise, we use mixed precision, where all heavy computation is done in bfloat16, whereas the optimizer state and weights are kept in full precision. To study the stability of our model, we experiment with training fully in reduced precision.

Our main result is a substantial speed-up of MoT models compared to dense Transformers (Figure 7) and results comparable to sparse MoEs (Figure 6). What follows is the analysis of the scaling properties of the MoT architecture with respect to the number of parameters (Figure 4) and the number of mixtures sent to each expert (Figure 5). We investigate the model's performance in low precision in order to simulate training instability and find that MoT is less susceptible to instabilities arising from low-precision training. Lastly, we show the connection between MoT and MoE, by spending an additional fraction of pretraining compute to effectively transform a MoT model into a Token Choice model (Section 4.4).

## 4.1 Model Architecture

The base of our experiments is a decoder-only Transformer based on GPT-2 [42]. We conduct experiments on two model scales: a 77M Medium model and a 162M Base model (refer to Appendix A for hyperparameters and training details). To obtain a Mixture of Tokens model, we replace the second half of the Feed-Forward layers in the Transformer with MoT layers. Because, similar to MoE models, the FLOPs and parameter counts in MoT are decoupled, we indicate the model architecture by its dense counterpart in terms of the number of FLOPs and, separately, the number of experts (or equivalently, group size). To this end, a MoT-Medium/32E model is one that uses the same number of FLOPs as a Medium (77M) Transformer model but uses 32 experts in MoT layers.

As outlined in Section 3.2, Medium/32E/4 signifies a model employing MoT layers with $32 \cdot 4$ small experts, which add up to the same number of parameters as 32 normal experts.

In addition to using the Transformer as a baseline, we also compare against Token Choice [27] and Expert Choice [30] as sparse MoE baselines. Given that Expert Choice is sensitive to the size of the batch, in order to avoid discrepancy between training and inference, we group tokens prior to routing in training Expert Choice models.

## 4.2 Scaling Results

Mixture of Tokens models demonstrate strong scaling properties with respect to the number of parameters. As seen in Figure 4, increasing the number of experts in MoT layers while using the same compute budget yields consistent improvements. All MoT models are a strict improvement over the Transformer. The figure also includes an ablation experiment, where the mixing weights are fixed to $1/n$, where $n$ is the group size. This corresponds to a uniform mixing strategy; the performance of that model clearly suffers, confirming that MoT layers learn non-trivial mixing strategies.

The increased number of token mixtures described in Section 3.2 represents another axis of scaling for MoT models, once again exhibiting consistent improvements. We hypothesize that this phenomenon is due to two mechanisms: First, the model becomes more expressive with a larger number of smaller experts. Second, the model can allocate its focus (the mixing weights) more flexibly to more important tokens while reducing the updates for trivial ones.

## 4.3 Comparison with the Transformer and Sparse MoEs

Crucially, the performance of Mixture of Tokens is comparable to that of the strong Mixture of Experts baselines (Figure 6). An increased number of mixtures allows it to compete with both Expert Choice and Token Choice architectures. As the sparse routing is hypothesized to contribute to training instabilities in large sparse models, Mixture of Tokens, being continuous, presents a promising

---

[3]https://huggingface.co/datasets/c4, dataset licensed under ODC-By.

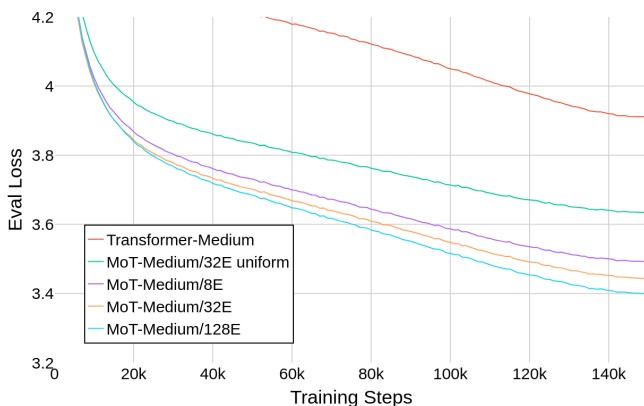

Figure 4: Scaling with respect to the number of parameters. Also featured are the Transformer baseline and an MoT model with a non-learnable, uniform routing strategy.

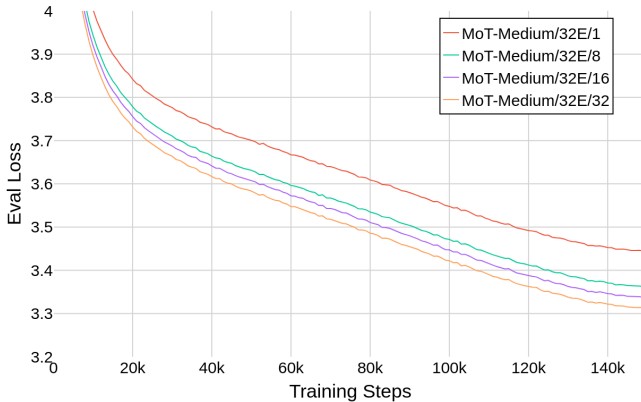

Figure 5: Scaling with respect to the number of token mixtures.

alternative. To investigate training instabilities at the scale we experiment on, we trained models entirely in bfloat16, as opposed to the mixed precision used in all other experiments. The results confirm that MoT is more resistant to lower precision training: as the precision of training decreases, the performance of Expert Choice drops below that of Mixture of Tokens, despite the former attaining better perplexity using mixed precision. We find this to be evidence of the architecture's potential for stable training at higher model scales. See Table 1 for details.

Finally, we combine our findings on MoT scaling properties to train our most efficient MoT model and compare it to the Transformer baseline (Figure 7). The result is a model that achieves the final loss of the baseline in one-third of the training steps. This represents a $3\times$ improvement in terms of the compute budget.

Table 1: Comparison of training result loss comparison. MoT performs better in the bfloat16-only setting. Learning rates were separately tuned in lower precision for both EC and MoT. Results are averaged over 3 random seeds.

|  | MoT-Medium/32E | Expert Choice-Medium/32E |
| --- | --- | --- |
| Mixed Precision | 3.442 ($\pm$ 0.002) | **3.420 ($\pm$ 0.002)** |
| bf16 only | **3.661 ($\pm$ 0.007)** | 3.728 ($\pm$ 0.044) |

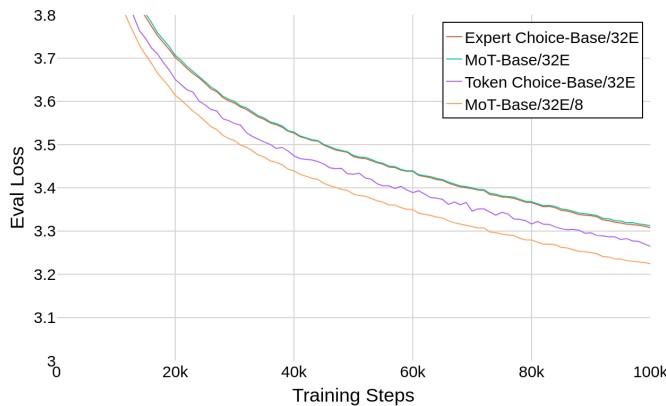

Figure 6: Comparison of MoT and sMoE architectures. An increased number of smaller experts allows MoT to match the performance of the best sMoE model. Due to computational constraints, the models were trained for 100K steps.

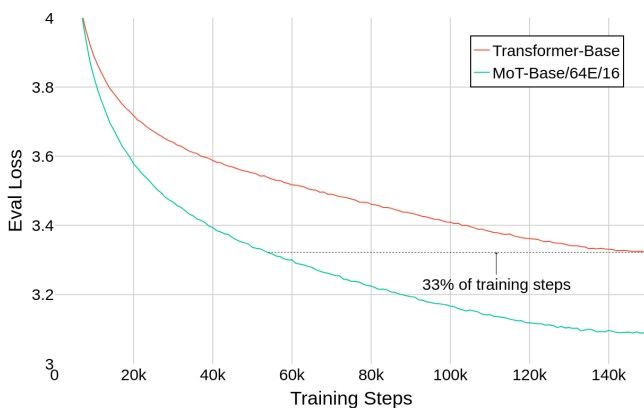

Figure 7: Our best MoT model reaches the final loss of the baseline in just 33% of the compute budget.

## 4.4 Transition Tuning

Mixture of Tokens suffers from a drawback common to MoEs, namely, it does not support unbatched inference. This is a direct consequence of its design - in the forward pass, it groups several tokens from different examples in the batch. With the growing adoption of Large Language Models on consumer hardware [43, 44], this lack of support could hinder the architecture's wider adoption. While a Mixture of Tokens with a group size of one is technically possible, in order to keep FLOPs constant, the layer would need to trivially reduce to a standard Transformer MLP.

To address this issue, we demonstrate that the weights learned by the Mixture of Tokens can be used to directly initialize a Token Choice model of the same specifications (number of experts and expert size). The layer responsible for producing mixing weights is utilized to initialize the sparse router. In order to mitigate the difference in performance that is caused by this change in architecture, we train the entire new model (no weights are frozen) for 10% of the total pretraining steps of the original model in order to recover the original model's performance (measured in eval loss). We call this technique *transition tuning*. This way, it is possible to train with Mixture of Tokens and enjoy unbatched generation at inference time. We hypothesize that this pipeline would be especially attractive in setups where having parts of the model train in higher precision is impossible, e.g., on specialized, low-precision hardware. The results are presented in Figure 8.

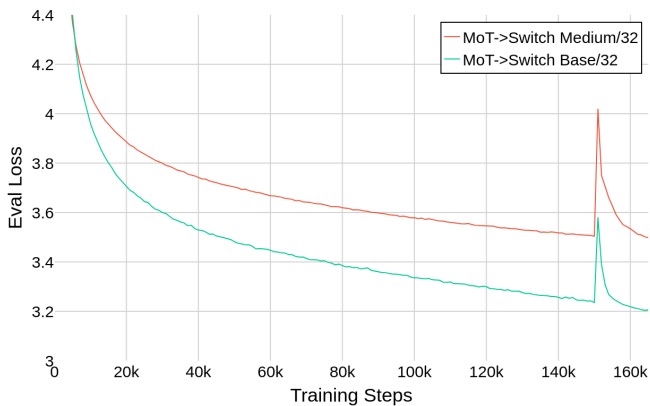

Figure 8: Transition tuning: The first $150$K steps of the model are completed using the Mixture of Tokens architecture. Then, a new Token Choice model is initialized with weights from the MoT model, and the model trains for an additional $15$K steps to recover performance. The spike in loss results from the sudden change of architecture.

## 5   Limitations and Future Work

With the strong performance of MoT on medium-sized models, an obvious next step is to train larger models. This would present an opportunity to validate the stability results on larger models, where training instabilities are more common.

As with most Mixture of Experts models, the memory footprint of MoT layers is substantial. Scaled models require large amounts of RAM on specialized hardware for training, making their adoption expensive. To this end, an attractive future direction would be to investigate model distillation with Mixture of Tokens models.

In this work, we experimented only with text modality in an autoregressive manner. Other modalities, such as vision, heavily overlap with the approach presented in work concurrent to ours [15].

Lastly, both training and inference with MoT involve mixing different examples within a single batch. This mixing of tokens from different sequences and the requirement of performing batched inference may be undesirable in some use cases. While performing unbatched inference is always inefficient with LLMs, as the memory throughput to access model weights becomes the bottleneck, unbatched inference still finds its uses. Even though transition tuning solves this problem, exploring different inference strategies might bring new insights.

## 6   Conclusions

In this work, we presented the Mixture of Tokens, a novel continuous Mixture of Experts architecture compatible with autoregressive decoding. This architecture scales to model sizes similar to sparse Mixture of Experts models, matches their performance, and is more resistant to training instabilities due to lower precision training. Moreover, we introduced transition tuning, a technique for initializing an MoE model with another pretrained MoE model of a different architecture, and showed that the new model achieves the performance of the original one using a fraction of the compute budget.

## Acknowledgements

We would like to express our sincere gratitude to Piotr Padlewski and Tomasz Trzciński for their general feedback and Dagmara Rudzińska for her invaluable support with graphic design and to Piotr Sankowski for providing an excellent scientific environment.

This work was funded by IDEAS NCBR, which also provided significant computational resources. Marek Cygan was partially supported by an NCBiR grant POIR.01.01.01-00-0392/17-00. The research was supported by PL-Grid infrastructure (grants PLG/2023/016148, PLG/2024/017060). We also benefited from the Entropy cluster (hosted at the Faculty of Mathematics, Informatics, and Mechanics of the University of Warsaw) funded by NVIDIA, Intel, the Polish National Science Center grants UMO-2017/26/E/ST6/00622 and 2022/45/N/ST6/02222, and ERC Starting Grant TOTAL. Part of the evaluations was done using computational resources provided by Writer.

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

# A    Training Hyperparameters

All models were trained using mixed precision unless explicitly stated otherwise. We conducted all experiments with a batch size of 256 and a context length of 256 for 150K training steps (unless explicitly stated), resulting in a total of 10B training tokens. We used the AdamW optimizer with default hyperparameters. When necessary, we adopted a Fully Sharded Data Parallel approach from PyTorch to parallelize training across multiple machines. Learning rates were tuned separately based on model size and architecture. The optimal learning rate for Transformers was 1e-3 for Medium models and 4e-4 for Base models, while for both MoT and MoE, they were 7e-4 for Medium models and 2e-4 for Base models.

Table 2: Training hyperparameters. The table provides example models featured in our experiments. All remaining models can be derived from this table.

| Model | Experts | Expert size | Group size | Total params | Blocks | $d_{model}$ | $d_{ff}$ | #att. heads |
|---|---|---|---|---|---|---|---|---|
| Transformer-Medium | - | - | - | 77M | 8 | 512 | 2048 | 8 |
| MoT-Medium/32E | 32 | 2048 | 32 | 336M | 8 | 512 | - | 8 |
| MoT-Medium/32E/8 | 256 | 256 | 32 | 337M | 8 | 512 | - | 8 |
| Transformer-Base | - | - | - | 162M | 12 | 768 | 3072 | 12 |
| MoT-Base/32E | 32 | 3072 | 32 | 520M | 12 | 768 | - | 12 |
| MoT-Base/64E/16 | 1024 | 192 | 64 | 977M | 12 | 768 | - | 12 |

# B    Downstream Evaluation

When trying to predict how specific changes to the model architecture will impact large-scale models, comparing perplexity can provide a reliable indication of model improvements. However, for completeness, we also measured performance on several downstream tasks relevant at this model scale, comparing MoT-Medium to Transformer-Medium, without fine-tuning, in a zero-shot setting. In these evaluations, for MoT, a single evaluation query is included in a batch of 32, with the remainder of the batch comprised of random sequences from the C4 training dataset, ensuring it remains zero-shot. We observe predictable improvements with the Mixture of Tokens on tasks PIQA [45], HellaSwag [46], and ARC-e [47], see Table 3.

Table 3: Performance of a medium-sized model on downstream benchmarks.

| | Transformer-Medium | MoT-Medium/32E/1 | MoT-Medium/32E/16 |
|---|---|---|---|
| PIQA | 60.2 | 62.4 | **65.8** |
| HellaSwag | 27.3 | 31.1 | **33.3** |
| ARC-e | 35.5 | 37.3 | **39.6** |

# C    Reproducibility

The code and configuration files used to produce the results described in this work are available in our public repository at `https://github.com/llm-random/llm-random`.

# D    Socio-Economic Impacts

The goal of this paper is to advance the field of Transformer training and Machine Learning in general, language modeling in particular. With Large Language Models exhibiting the most impressive results in Machine Learning to date, we believe that this work is able to help advance the model capabilities even further. As to the potential socio-economic consequences of our work, it shares the common potential impact of all work done on training efficiency.

# E  Compute Resources

Table 4: Compute resources used for each experiment. All models were trained on NVIDIA A100 GPUs, with either 40 or 80 GB of RAM.

| Model | GPU RAM | Time | GPUs |
|---|---|---|---|
| Transformer-Base | 40GB | 32h 20m | 1 |
| MoT-Base/64E/16 | 40GB | 33h 12m | 2 |
| MoT-Medium/128E | 40GB | 26h 36m | 1 |
| MoT-Medium/32E | 40GB | 23h 9m | 1 |
| MoT-Medium/8E | 40GB | 22h 31m | 1 |
| MoT-Medium/32E | 40GB | 20h 17m | 1 |
| Transformer-Medium | 40GB | 18h 48m | 1 |
| MoT->Switch Medium/32 | 40GB | 35h 11m | 1 |
| MoT->Switch Base/32 | 40GB | 17h 20m | 4 |
| MoT-Medium/32E/1 | 40GB | 23h 9m | 1 |
| MoT-Medium/32E/8 | 40GB | 24h 13m | 1 |
| MoT-Medium/32E/16 | 40GB | 25h 36m | 1 |
| MoT-Medium/32E/32 | 40GB | 28h 20m | 1 |
| Expert Choice-Base/32E | 80GB | 21h 12m | 2 |
| MoT-Base/32E | 80GB | 19h 38m | 2 |
| MoT-Base/32E/8 | 40GB | 22h 10m | 2 |
| Token Choice-Base/32E | 80GB | 20h 19m | 8 |

# F  Contributions

Szymon implemented a PoC and different variants of MoT, together with running experiments and optimizations. Michał implemented and experimented with various MoT designs and contributed to the infrastructure design and implementation. Sebastian provided the initial idea, research intuitions, and direct project supervision. Maciej was responsible for parts of evaluation and significant engineering. Jakub implemented MoE baselines, Jan stabilized Mixture of Experts training, while both helped with MoE hyperparameter tuning. Tomasz consulted ideas and helped with cluster infrastructure. Kamil and Krystian contributed to general engineering. Everybody above contributed to the infrastructure of the project. Marek provided scientific advice and high-level supervision.

