# OpenReview forum: "Mixture of Tokens: Continuous MoE through Cross-Example Aggregation"
_NeurIPS.cc/2024/Conference — NeurIPS 2024 poster_

### Official Review · Reviewer_ef5W · 2024-07-03

**Soundness:** 3
**Presentation:** 3
**Contribution:** 3
**Rating:** 6
**Confidence:** 4

**Summary:**

This paper proposes a new MoE architecture called Mixture of Tokens (MoT). The motivation for this architecture is twofold: first it is the training instabilities incurred (among other things) by low precision training in standard sparse MoEs; secondly, it is the discontinuous nature of sparse MoEs that makes them hard to train. This work seem to be highly inspired by [1].

In contrast to MoEs, which select top-k subset of experts for each token, MoT aggregates tokens within groups created by subsetting examples across the batch dimension. The cross-example token aggregation weights are produced by a router conditioned on the tokens in the corresponding group. The aggregated group representations are then passed through all the experts. The output token representation is produced by a weighted average of all experts' outputs for a group, where the weights are the original aggregation weights.

The paper measures validation perplexity of MoT comparing it to compute matched dense transformer and sparse MoEs demonstrating its superiority over the prior and on par performance with the later. MoT shows better training stability in low precision training regime as compared to sparse MoEs.

[1] Puigcerver, Joan, et al. "From sparse to soft mixtures of experts." arXiv preprint arXiv:2308.00951 (2023).

**Strengths:**

Overall, the paper proposes in interesting and novel extension of soft MoE to autoregressive problems. The idea of mixing tokens across examples in the batch is very interesting, given that it does not break the autoregressive property.

Originality: good. The idea of soft MoEs is not new, but its adoption to autoregressive problems is novel to the best of my knowledge.

Quality: The claims seem to be supported by evidences, yet experimental evaluation might be insufficient (see questions/weaknesses section).

Clarity: the paper is well organized. In terms of writing, several formulations might need to revisited (see weaknesses/questions).

Significance: the paper makes a moderate contribution to the field and can potentially impacting future research.

**Weaknesses:**

One weakness of this paper is a lack of intuitive explanation of why mixing across examples makes sense? I understand that it results in more stable training due to lack of discreet operations, but why does it make sense from the modelling perspective?

If sentences in the batches are consecutive interrelated text passages, then MoT effectively increases the context length by letting the model to also rely on cross-example information. Can this have an effect on generalization ability? e.g. in instruction fine-tuning, examples in the batches are usually not related to each other, hence MoT type of pre-training may be detrimental for downstream task adaptation of these models, where downstream tasks examples are completely unrelated.

Therefore, another potential weakness for me is a lack of  generalization/downstream adaptation analysis. Does the training stability advantage of MoT improves generalization and adaptability of the model to downstream tasks? Or does cross-example information sharing actually weakens downstream task generalization?

**Questions:**

- ll. 52 - 61: I am uncertain if points 2 and 4 qualify as contributions of this work. These appear to be properties of the proposed MoT method rather than contributions in themselves.
- ll. 55-57: If I understand correctly, the key comparison should be with other sparse MoE methods rather than dense transformers. Therefore, why emphasize a speedup over dense transformers as a contribution when the more relevant comparison is the speedup over other sparse MoE architectures?
- l. 104: "Mixture of Experts, introduced into deep learning by [24],..." does not seem to be correct. MoEs have a much longer history, please consider works [2,3];
- l. 190: why half of feed-forward layers are replaced with MoT layers?
- ll. 167 - 169: an important difference to [15] that might be mentioned explicitly here, is that MoT aggregates tokens across examples in a batch, while [15] aggregates patches within each example.
- caption Figure 6: if I am not mistaken, MoT here not only matches but outperforms best sMoE
- since one of the main advantages of the proposed MoT is that it does not require additional load balancing tricks etc., it would be interesting to include a empirical comparison (or at least a discussion) to existing sparse MoEs that also do not require such tricks, see e.g. [4]
- how are examples in the batches created? Are these consecutive interrelated text passages from the same underlying text?
- MoT's weighted token grouping is reminiscent of an attention operation, but across examples in a batch. Is there any intuition for why aggregating information across examples can be useful? One potential way it can be useful, is that if sentences in the batches are related (e.g. maybe these are consecutive sentences from the underlying training text) then of course  incorporating information from previous/future examples can help reducing loss (training and validation), since the examples (samples in a batch) are interrelated.

Overall, I believe this paper has significant potential. However, I have concerns regarding downstream task generalization and the relationship between the examples in the training batches. I am willing to reconsider my rating upon rebuttal.

[2] Robert A Jacobs, Michael I Jordan, Steven J Nowlan, and Geoffrey E Hinton. Adaptive mixtures of local experts. Neural computation, 3(1):79–87, 1991b.

[3] Robert A Jacobs, Michael I Jordan, and Andrew G Barto. Task decomposition through competition in a modular connectionist architecture: The what and where vision tasks. Cognitive science, 15(2):219–250, 1991a

[4] Csordás, Róbert, Kazuki Irie, and Jürgen Schmidhuber. "Approximating two-layer feedforward networks for efficient transformers." arXiv preprint arXiv:2310.10837 (2023).

**Limitations:**

Limitations are addressed in Sec. 5.
I think one missing limitation of this work is the fact that no downstream task generalization is evaluated.

---

> ### Author Rebuttal · Authors · 2024-08-07
>
> We would like to thank the reviewer for their questions and suggestions and appreciate the recognition of our paper's strengths. Below, we address the weaknesses and questions mentioned in the review. If our answers address the reviewer's concerns, we would like to kindly ask for the reconsideration of the rating.
>
> **Regarding previous related work.** We would like to refer to our reply in the general comment.
>
> ## Regarding weaknesses:
> **W1:** We agree that, apart from experimental results, an intuition behind mixing tokens across examples should be provided. We will add such an explanation to the camera-ready version.
>
> As we mix tokens from multiple unrelated sequences, we do not expect the model to meaningfully use the information from one sequence to improve prediction in a different sequence. However, we argue that such mixing (1) provides richer feedback (gradients) to train the model, especially the router, and (2) provides a smoother loss landscape, resistant to small perturbations in inputs/weights.
>
> Intuitively, for a given expert, **from the perspective of each token**, the token gets some amount of update to its representation (in the residual stream) based on:
> 1.  Itself, which produces a proper signal expected to improve the token representation.
> 2. Tokens other than itself, which are essentially random tokens from unrelated sequences. As those sequences are randomly sampled from the dataset, the impact from those tokens will point in random directions and, essentially, just add some amount of noise to the token update.
>
> We generally expect neural networks to be resistant to some amount of noise added to them (e.g. dropout doesn't hurt). Moreover, while the signal-to-noise ratio worsens for tokens with low expert weight, the expert weight also modulates the magnitude of the update. Therefore, the amount of noise added to the representation is limited.
>
> We stipulate that MoT experts will learn to focus on a single token or a small number of tokens - minimizing the noise and approximating sparse MoE when optimal. Still, other tokens will be assigned nonzero weight, enabling some information to flow to the router for each and every token-expert pair (in contrast to sparse MoE). Moreover, the output of MoT is more continuous, with small perturbations of input/weights corresponding to small changes in the output instead of large discrete jumps with sparse MoE.
>
> **W2:** During the training and evaluation, we essentially put  random, unrelated examples into the same group. For reasons explained in **W1** not much cross-example information transfer occurs. Therefore, it does not have any effect on generalization ability.
>
> **W3:** We believe that comparing perplexity generally predicts model improvements more reliably, especially when trying to predict how particular changes to the model architecture will impact extremely large-scale models. However, during the short rebuttal period, we measured performance on a few benchmarks relevant at this model scale, comparing MoT-Medium to Transformer-Medium, without fine-tuning, zero-shot. In these evaluations, for MoT, just a single evaluation query is put into a batch of 32, with the rest of the batch being comprised of random sequences from the C4 training dataset - ensuring this is still zero-shot.
>
> | Metric | Dense Transformer | MoT/32E/1 | MoT/32E/16 |
> | ------ | ----------------- | ----------------- | ------- |
> | PIQA | 60.2 | 62.4 | **65.8** |
> | HellaSwag | 27.3 | 31.1 | **33.3** |
> | ARC-e | 35.5 | 37.3 | **39.6** |
>
> ## Regarding questions:
> **Q1:** After some consideration, we agree with the reviewer, and we will move the results/properties of MoT itself (in particular, speed-up against the dense model and improved stability compared to MoE) from the contributions bullet points to the main introduction part.
>
> **Q2:** We think it is important to compare MoT to both kinds of models—dense models, where the main benefit we show is a significant speedup (point 2), and MoE models, where the main benefit we focus on is increased stability of the training (point 4). We will make those points clearer while moving them into the main text.
>
> **Q3:** Thank you for bringing this to our attention. This is an oversight on our part. While we already cite [5] (in line 75), we will change the citation in line 104 as well (to both [5] and [6]), to ensure proper credit is given.
>
> **Q4:** In the literature on MoE (e.g. [1][7][8]), it is quite standard for MoE layers to replace only half of FFN layers - this keeps the majority of MoE advantages while requiring significantly fewer total parameters.
>
> **Q5:** We will revise this paragraph to highlight the differences and similarities between SoftMoE and MoT more clearly, as suggested.
>
> **Q6:** While MoT indeed outperforms sparse MoE here, the differences between the compared approaches are relatively minor. Given these results, we think it would be an exaggeration to claim that our method outperformed the others. Instead, we focus on showing other advantages of MoT, like increased training stability.
>
> **Q7:** Please note that one of the methods we compare our approach to is the expert-choice variant of MoE that operates without load-balancing. However, it employs a sparse router, which requires higher precision and is not fully differentiable, leading to unstable training. By replacing the sparse router with our fully differentiable counterpart, we achieve stable training and enable training entirely in lower precision.
>
> **Q8:** Training batches are created in the same manner as for regular training. We sample random passages from the training dataset until the desired number of examples in the batch is reached. Consequently, it is unlikely that any two passages within a batch are related.
>
> **Q9:** See our explanation of intuition behind MoT in the previous section **W1** of this response.

---

> > ### Comment · Reviewer_ef5W · 2024-08-07
> >
> > I want to thank the authors for their elaborate replies! I will increase my score.

---

> > > ### Author Response · Authors · 2024-08-09
> > >
> > > We appreciate the reviewer's engagement and we are thankful for the update of the score.

---

### Official Review · Reviewer_wt4H · 2024-07-12

**Soundness:** 3
**Presentation:** 2
**Contribution:** 2
**Rating:** 6
**Confidence:** 3

**Summary:**

In this paper, the authors propose a novel method called Mixture of Tokens (MOT), an expert-based architecture that addresses the drawbacks of existing Mixture of Experts (MoE) approaches, such as reduced training efficiency, instability, and the necessity of using load balancing losses and elaborate training schedules. These issues are often caused by the top-k operation used to select the active experts. The authors' approach combines tokens into groups (subsets of several tokens) and uses a weighted sum of tokens within these groups as input to each expert, with each expert using its own weights for the sum. This allows each token within the group to be processed by each expert, resolving the issues associated with top-k routing. The authors validate their method through extensive experiments on an autoregressive language modeling task, demonstrating significant improvements in training stability, reduced training costs (compute budget), and improved scaling with regard to model size.

**Strengths:**

* The paper is clearly written and easy to follow. The idea is intuitive and easy to grasp. The related work section provides an adequate discussion of existing approaches (MoE and variations), highlights their problems, and explains the way to resolve them

* Based on the weaknesses of existing MoE approaches, the authors develop an efficient MoT approach that is relatively simple to integrate into existing architectures, does not increase computational cost, and provides important advantages such as more stable training and better convergence

* The authors provided an extensive evaluation of their approach, showing a 3x training speedup compared to common MoE architectures and improved performance in several important setups (such as low-precision training). Therefore, I find the contribution and value of the proposed approach to be clear and evident.

**Weaknesses:**

* One of the weaknesses the authors pointed out in the limitations section is the necessity to use batched inference for the approach, which could limit the applicability scope of the method. While there are still important applications for batched inference, I believe that resolving this issue could significantly increase the influence of the method

* From the perspective of the experimental evaluation, I would be curious to see evidence that the behavior demonstrated in the paper would hold in other domains, such as images. Additionally, it seems that the proposed approach could resemble the method in [1] a lot.

* In terms of experiments, it would also be interesting to see comparisons with other MoE approaches mentioned in the related work, beyond the reported Token Choice and Expert Choice, or more novel approaches such as [2] and similar methods.

[1] Puigcerver, Joan, et al. "From sparse to soft mixtures of experts." ICLR 2024
[2] Anthony, Quentin, et al. "Blackmamba: Mixture of experts for state-space models." ICLR 2024

**Questions:**

* As stated in the weaknesses section, is it possible to demonstrate the effectiveness of the approach beyond the autoregressive language modeling task? I believe it could significantly strengthen the paper.

* In addition to the previous question, could you elaborate on what the major differences are between the proposed approach and the [1] method? Is it the fact that in MoT mixing happens within different batch inputs and in [1] we mix tokens within the same image? If so, then the novelty of MoT (compared to [1]) comes from the grouping and the application to the autoregressive task, which could undermine the contribution of the paper.

[1] Puigcerver, Joan, et al. "From sparse to soft mixtures of experts." ICLR 2024

**Limitations:**

No significant limitations of the paper, though the authors' discussion on limitations is much appreciated.

---

> ### Author Rebuttal · Authors · 2024-08-07
>
> We want to thank the reviewer for their comments and questions. We also appreciate the mention of the simplicity of integrating our method into existing approaches. If the reviewer's concerns have been addressed, we would like to kindly ask for the reconsideration of the rating.
>
> **Regarding batched inference and applicability.** We agree that the necessity of using batched inference, while fine for many industry applications, is a limitation for others. We would like to highlight, however, that transition tuning (introduced in Section 4.4 of our work) can lift this limitation by enabling many benefits of MoT during training (e.g. increased stability) while converting the model to sparse MoE for unbatched inference.
>
> **Regarding other domains and tasks other than autoregressive language modeling.**  We agree that testing the method on other domains is an interesting avenue of research. With limited time and budget, we focused on the domain with, arguably, the most pressing problem of efficiency because of recent scaling efforts across the whole industry. Apart from text the important domain to test MoT on would be vision - but those experiments would heavily overlap with a concurrent paper of SoftMoE. We will properly mention that in the Limitations section of our work.
>
> **Regarding the differences with SoftMoE.** As the reviewer noted, the main difference between MoT and SoftMoE is that MoT mixing happens between different sequences and not within a sequence/image, enabling its use on autoregressive tasks. We would like to refer to our reply in the general comment for further information about concurrency and the novelty of our work compared to SoftMoE.

---

> > ### Comment · Reviewer_wt4H · 2024-08-12
> >
> > Thank you for your detailed rebuttal and for addressing the questions raised. I appreciate the clarifications provided, particularly regarding batched inference and the differences between MoT and SoftMoE. Given the solid contributions and your thorough responses, I am maintaining my original rating.

---

> > > ### Author Response · Authors · 2024-08-12
> > >
> > > We thank the reviewer for their response. We are glad that we were able to clarify the points raised.

---

### Official Review · Reviewer_BdV2 · 2024-07-13

**Soundness:** 3
**Presentation:** 2
**Contribution:** 2
**Rating:** 6
**Confidence:** 4

**Summary:**

This paper proposes a new routing algorithms for MoEs: the mixture of tokens. The context is the following: routing in MoEs is tricky because one token gets usually assigned to one or a few experts and so the gradient feedback to update the router is not great. Therefore, several recent papers proposed to either average the experts parameters [1,2] or mixing the tokens [3] to overcome this issue. This paper focuses on this research direction. Prior work [3] did it in the case of computer vision where causality is not an issue. This paper tries to adapt mixture of tokens to the case of autoregressive models. They first explain how they do the mixture of tokens in section 3.1 and then detail how they manage to ensure causality in section 3.2. Lastly in section 4, they show the benefits of their method over other routing algorithms: in particular, they show that Mixture of Tokens (MoT) minimizes the eval loss much faster than dense Transformers and the performance is slightly better than standard MoEs with token choice and expert choice routings.






[1] Zhong, Z., Xia, M., Chen, D., & Lewis, M. (2024). Lory: Fully Differentiable Mixture-of-Experts for Autoregressive Language Model Pre-training. arXiv preprint arXiv:2405.03133.
[2] Muqeeth, M., Liu, H., & Raffel, C. (2023). Soft merging of experts with adaptive routing. arXiv preprint arXiv:2306.03745.
[3] Puigcerver, J., Riquelme, C., Mustafa, B., & Houlsby, N. (2023). From sparse to soft mixtures of experts. arXiv preprint arXiv:2308.00951.

**Strengths:**

I think that mixture of tokens is an idea that has been around for a while [1] and i was curious to know how it behaves in the context of autoregressive models. So in this aspect, I think the paper is interesting. Also, I found the method well-presented.




[1] Puigcerver, J., Riquelme, C., Mustafa, B., & Houlsby, N. (2023). From sparse to soft mixtures of experts. arXiv preprint arXiv:2308.00951.

**Weaknesses:**

In general, I am a bit doubtful about the methods that attempt to merge either tokens or parameters in order to solve the ill-posed problem of routing in MoEs. For me, MoEs are primarily introduced for efficiency in that we can increase the size of the model while keeping the same number of FLOPs. In my opinion, the approaches that merge tokens/parameters lose the efficiency aspect. I think the authors tried to alleviate this issue by using smaller experts (as done in [1]) but I don't think this totally solves the problem. [1] also shows that when the granularity is too high, MoEs performance drop. In any case, I am happy that some researchers tried mixing tokens in the context of autoregressive models but I do not fully believe in it. Here are some more precise questions I would like to ask to the authors:

- **Computational cost**:Can the authors analyze the additional number of FLOPs that their method incurs compared to standard token choice? Or to other merging methods like the parameter merging in [2]?

- **Any advantage over expert choice/token choice?**: do you think there is any benefit from using mixture of tokens over other existing routing algorithms? It is not obvious from Figure 6. I understand the primary motivation and it makes sense to me. But in practice, it does not seem that the merging methods yield any significant benefits over token or expert choice.

- **Preserving the causal structure**: This is always a challenge for routing algorithms like expert choice or Lory that do not naturally preserve causality. Have you tried other schemes to preserve the causal structure? Would the one that is used when running expert choice on autoregressive models fail? Can you clarify why you believe there is no information leakage, sorry I may have missed this point?


[1] Krajewski, J., Ludziejewski, J., Adamczewski, K., Pióro, M., Krutul, M., Antoniak, S., ... & Jaszczur, S. (2024). Scaling laws for fine-grained mixture of experts. arXiv preprint arXiv:2402.07871.
[2] Zhong, Z., Xia, M., Chen, D., & Lewis, M. (2024). Lory: Fully Differentiable Mixture-of-Experts for Autoregressive Language Model Pre-training. arXiv preprint arXiv:2405.03133.

**Questions:**

I have written my questions above.

**Limitations:**

The authors have clearly mentioned the limitations of their approach.

---

> ### Author Rebuttal · Authors · 2024-08-07
>
> We would like to thank the reviewer for their feedback and questions. We also appreciate the recognition of the value of the research problem and the presentation of our work. We hope that the answers below adequately answered the reviewer's questions. If that is the case, we kindly ask for a reconsideration of the paper score.
>
> **Regarding SoftMoE.**
> We would like to refer to our reply in the general comment.
>
> **Regarding computational cost.** The time complexity of our approach doesn't change compared to standard token choice or expert choice methods. The cost of computing routing logits itself takes $O(d_{model}*N_{experts}*N_{tokens})$ FLOPs, and this time complexity is the same for all MoE variants. The mixing and unmixing operations in MoT have the same time complexity. Regarding granularity, our models use values around compute optimal for this model size as calculated in [3]; therefore, according to their experiments, performance should not drop, as in the case of extreme granularity. We will clarify this in the camera-ready version and add a citation to their work.
>
> **Regarding the benefits of MoT over expert choice/token choice.** We believe that MoT and techniques developed in the future based on MoT will provide better training stability (shown in our work), and therefore enable improved training performance, especially at scale. Moreover, within a continuous setting, each expert is trained on every token, which might be beneficial once we scale to higher expansion rates.
>
> **Regarding the causal structure.** The original expert choice routing did not try to preserve the causal structure by default, with the Limitations section in [4] stating, "The expert choice method might not immediately apply to auto-regressive text generation as our current implementation takes in the past and future tokens to perform the top-k selection." The approach used in our paper is, we think, the only natural adaptation of expert choice to autoregressive models preventing information leaks. This method is used both in our expert-choice and MoT models. The approach of preventing the causal information leak used in [2] is not applicable to MoT. In MoT, we can never mix tokens from the same sequence, and routing based on the previous tokens/segment (rather than the current tokens/segment) doesn't mitigate the issue.
>
> **Regarding no information leakage in MoT.** In MoT, there is no information leakage, as we never mix tokens from the same sequence together. Each mixture is based on a specific position in a sequence. So, a token at position $i$ in a sequence $s$ can, in a MoT layer, see tokens from any sequence at position $i$. Combined with the causal attention layer, token at position $i$ can see any tokens at positions $0:i$ in a sequence $s$ (as in standard causal Transformer), and also any tokens at positions $0:i$ in any sequence in a group/batch (through a combination of MoT layer being able to look at a different sequence, and causal attention being able to look at any previous position). However, looking at different sequences in a batch doesn't constitute an information leak since those other sequences are IID examples randomly drawn from the dataset.

---

> > ### Comment · Reviewer_BdV2 · 2024-08-09
> >
> > I would like to thank the reviewers for their reply. I appreciate their clarification regarding the computational cost and I believe that this should be added to the final version of the paper. I increase my score by 1 point.

---

> > > ### Author Response · Authors · 2024-08-12
> > >
> > > We appreciate the reviewer's suggestion and are grateful for the improved score. We will include a clarification on the computational cost in the camera-ready version of the paper.

---

### Author Rebuttal · Authors · 2024-08-07

We thank all the reviewers for assessing our paper. We appreciate all the positive comments and will consider all the feedback to improve our work.

We address each review's comments and questions in individual responses. In this general reply, we want to address just the shared concerns about comparison, concurrency, and novelty between MoT and SoftMoE[1]. Finally, we also provide references, which are shared among all of our replies.

**Regarding SoftMoE[1].** We would like to highlight that, as we stated in Section 3.3, our method was developed independently and concurrently with SoftMoE [1]. Many of our experiments were conducted and MoT code was publicly accessible before SoftMoE was published  (due to anonymization we cannot provide links here; we can verify with AC if requested).

While we think that the concurrency of our work and SoftMoE is enough to defend our contribution - we also would like to note the importance of developing this kind of method for language modeling. This is clear to see in the ICLR 2024 review process of SoftMoE, where the most common criticism is its limitation to only non-autoregressive tasks. This weakness is mentioned in 3 out of 4 reviews of their work, and it is the only "Justification For Why Not Higher Score" identified in a meta-review. This shows the importance of our work even without consideration of the concurrence of [1] and MoT.

# References (common for all replies):

[1] Puigcerver, Joan, et al. "From sparse to soft mixtures of experts." arXiv preprint arXiv:2308.00951 (2023).

[2] Zhong, Zexuan, et al.  "Lory: Fully Differentiable Mixture-of-Experts for Autoregressive Language Model Pre-training" arXiv preprint arXiv:2405.03133 (2024)

[3] Krajewski, Jakub, et al. (2024). "Scaling laws for fine-grained mixture of experts." arXiv preprint arXiv:2402.07871 (2024)

[4] Zhou, Yanqi, et al. "Mixture-of-Experts with Expert Choice Routing" arXiv preprint arXiv:2202.09368 (2022)

[5] Jacobs, Robert A, et al. "Adaptive mixtures of local experts." Neural computation, 3(1):79–87, 1991b

[6] Jacobs, Robert A, et al. "Task decomposition through competition in a modular connectionist architecture: The what and where vision tasks." Cognitive science, 15(2):219–250, 1991a

[7] Fedus, William, et al. "Switch Transformers: Scaling to Trillion Parameter Models with Simple and Efficient Sparsity." arXiv preprint  arXiv:2101.03961 (2022).

[8] Lepikhin, Dmitry, et al. "GShard: Scaling Giant Models with Conditional Computation and Automatic Sharding"  arXiv preprint arXiv:2006.16668 (2020).

---

### Decision · Program_Chairs · 2024-09-25

**Decision:**

Accept (poster)

**Comment:**

The paper proposes an interesting approach to MoE-style autoregressive modeling using soft combinations of tokens along the batch dimension. The reviewers unanimously liked the idea and found the paper easy to follow. Concerns about computational cost and downstream evaluations were addressed in the rebuttal. I feel that for future work, more extensive downstream evaluation of this architecture would be interesting, however, in accordance with the reviewers I believe that the paper & rebuttal provide sufficient evidence in their current state to justify publication. Congratulations on the acceptance!